# Outcome Analysis of Treatment Modalities for Thoracic Sarcomas

**DOI:** 10.3390/cancers15215154

**Published:** 2023-10-26

**Authors:** Milos Sarvan, Harry Etienne, Lorenz Bankel, Michelle L. Brown, Didier Schneiter, Isabelle Opitz

**Affiliations:** 1Department of Plastic Surgery and Hand Surgery, University Hospital Zurich, 8091 Zurich, Switzerland; milos.sarvan@usz.ch; 2Department of Thoracic Surgery, University Hospital Zurich, 8091 Zurich, Switzerland; harry.etienne@usz.ch (H.E.); didier.schneiter@usz.ch (D.S.); 3Department of Medical Oncology, University Hospital Zurich, 8091 Zurich, Switzerland; lorenz.bankel@usz.ch; 4Department of Radiation Oncology, University Hospital Zurich, 8091 Zurich, Switzerland; michelleleanne.brown@usz.ch

**Keywords:** chest wall sarcoma, chest wall resection, multimodality treatment, radiotherapy, chemotherapy, chest wall reconstruction, overall survival, recurrence-free survival, outcome analysis

## Abstract

**Simple Summary:**

Primary chest wall sarcomas are rare tumors that require oncological and surgical therapy. Data about this entity remain scarce. The aim of our retrospective study was to assess the outcome of multimodal therapies. Over a period of 19 years, forty-four patients with chest wall sarcoma received surgery, radiotherapy, and/or chemotherapy. The overall survival after 5 years was 51.1%. There were no perioperative mortality cases. While the presence of metastasis and the tumor grade were identified as factors reducing survival chances, local tumor recurrence and the margins of surgical resection did not significantly influence survival outcomes.

**Abstract:**

Background: Primary chest wall sarcomas are a rare and heterogeneous group of chest wall tumors that require multimodal oncologic and surgical therapy. The aim of this study was to review our experience regarding the surgical treatment of chest wall sarcomas, evaluating the short- and long-term results. Methods: In this retrospective single-center study, patients who underwent surgery for soft tissue and bone sarcoma of the chest wall between 1999 and 2018 were included. We analyzed the oncologic and surgical outcomes of chest wall resections and reconstructions, assessing overall and recurrence-free survival and the associated clinical factors. Results: In total, 44 patients underwent chest wall resection for primary chest wall sarcoma, of which 18 (41%) received surgery only, 10 (23%) received additional chemoradiotherapy, 7% (3) received surgery with chemotherapy, and 30% (13) received radiotherapy in addition to surgery. No perioperative mortality occurred. Five-year overall survival was 51.5% (CI 95%: 36.1–73.4%), and median overall survival was 1973 days (CI 95% 1461; -). As determined in the univariate analysis, the presence of metastasis upon admission and tumor grade were significantly associated with shorter survival (*p* = 0.037 and *p* < 0.01, respectively). Five-year recurrence-free survival was 71.5% (95% CI 57.6%; 88.7%). Tumor resection margins and metastatic disease upon diagnosis were significantly associated with recurrence-free survival (*p* < 0.01 and *p* < 0.01, respectively). Conclusion: Surgical therapy is the cornerstone of the treatment of chest wall sarcomas and can be performed safely. Metastasis and high tumor grade have a negative influence on overall survival, while tumor margins and metastasis have a negative influence on local recurrence.

## 1. Introduction

Chest wall malignancies are rare entities that can be classified as primary malignancies (mostly represented by sarcomas) and secondary malignancies (mostly represented by non-small-cell lung cancer and breast cancer) [1,2]. Approximately 55% of primary malignant chest wall tumors originate from bone or cartilage, and the remaining 45% originate from soft tissues [3]. Primary chest wall sarcomas account for less than five percent of all thoracic neoplasms [4]. They are a heterogeneous group of malignancies that originate from mesenchymal cells. Surgical resection is part of the multimodality treatment recommended for these tumors. Radiotherapy is typically indicated for high-risk soft-tissue sarcomas—Grades 2–3—with neoadjuvant radiotherapy being used more frequently in recent years, extrapolating from the data for extremity sarcomas [5,6]. However, surgery is complex and requires close collaboration between plastic surgeons, thoracic surgeons, and/or orthopedic surgeons to achieve sufficient, safe surgical margins and reconstruct the large parietal defect that follows [7]. Indeed, prosthetic reconstruction and soft tissue coverage are often required to maintain chest wall stability, prevent flail chest, and preserve pulmonary function [8].

The early post-operative course is mainly marked by respiratory complications in a third of patients [9]. Long-term outcomes depend mainly on tumor histology and grade. No specific guidelines detail the type of muscle flap (pedicled or free flap) or prosthetic material that should be used. The aim of this study was to review our experience regarding the surgical treatment of chest wall sarcomas, evaluating the surgical and clinical short- and long-term results.

## 2. Materials and Methods

We undertook a retrospective, descriptive study conducted at the department of Thoracic Surgery of the University Hospital of Zurich. Patients who had been surgically treated for a chest wall sarcoma between 1999 and 2018 were included. Exclusion criteria were as follows:Patients with traumatic chest wall defects;Patients with congenital chest wall defects;Patients who had been surgically treated for chest wall malignancies other than sarcomas.

### 2.1. Ethical Statement

This study was approved by our institutional review board and was conducted in accordance with the declaration of Helsinki, and the protocol was approved by the Cantonal Ethic Committee Zurich (BASEC-Nr. 2020-00957). As per institutional protocol, prior informed and written consent was obtained from all the patients for future use of their data.

### 2.2. Patient Management

Tumor resectability and multimodal management were determined and overseen, respectively, by a dedicated multidisciplinary sarcoma board involving radiologists, thoracic oncologists, thoracic surgeons, radiation oncologists, orthopedics, and plastic surgeons. Preoperative work-up included a biopsy of the lesion, a thoracic computed tomography (CT) scan complemented by a chest Magnetic Resonance Imaging (MRI) scan, a positron emission tomography scan (PET-CT), lung function tests, and, if needed, cardiovascular testing. Neoadjuvant or adjuvant treatment modalities were dependent on type and stage of the sarcoma as well as resection margins. Whenever possible, in cases of high-risk or locally advanced soft tissue sarcomas, neoadjuvant chemotherapy was administered. Induction chemotherapy for osteosarcoma included a combination of doxorubicin, cisplatin, and ifosfamide, or the “EURO-E.W.I.N.G 99” study protocol with vincristine, ifosfamide, doxorubicin, and etoposide (VIDE) was used [10]. For soft tissue sarcoma, a dual chemotherapy regimen (e.g., doxorubicin and ifosfamide) was used. The surgical approach depended on the location, size, and depth of the tumor. The resection extent was set to at least 2 cm, if possible. Higher-grade sarcoma types received an extended resection margin of 4 cm. Thoracic wall reconstruction included implantation of mesh (e.g., polytetrafluoroethylene) alone or in combination with stable implants, such as titanium plates, depending on the size, number of ribs resected, and location. In the case of anticipated major skin defects after thoracic wall resection, plastic surgeons were involved to enable the preoperative planning of defect coverage. Sternal reconstructions required special materials (titanium cage (TRIONYX^®^, NEURO FRANCE Implants, Clamart, France)) and flap coverage. Preoperative or postoperative radiotherapy was performed by the Department of Radiotherapy in accordance with the sarcoma board. Radiotherapy was usually applied in 25–30 sessions with a cumulative dose of 50–60 Gy depending on whether a neoadjuvant or adjuvant application was required. Histological review was performed by dedicated musculo-skeletal pathologists to determine histology (soft tissue versus bony), grade, and staging.

### 2.3. Data Collection

Our hospital’s electronic records were retrospectively analyzed, and the following data were collected:Demographic characteristics (sex, age, comorbidities, American Association of Anesthesiologists (ASA) classification, and Karnofsky Index);Oncological therapy modalities;Details of surgical management using chest wall resection and reconstruction methods;Pathological features and resection margins;Perioperative morbidity and mortality;Long-term outcome.

Overall survival was defined as ranging from the date of operation to death or the date of the last follow-up for those patients who survived. Recurrence-free survival was defined as the period from the date of surgery to first local tumor recurrence. No variables were missing.

### 2.4. Assessment of Perioperative Complications

Complications were classified according to the Clavien–Dindo Classification grade. Overall postoperative outcome was determined using the validated “Comprehensive Complication Index” (CCI^®^). It sums all postoperative complications into a calculated value based on Clavien–Dindo grades, ranking the patients’ overall complications from 0 to 100 (death).

### 2.5. Outcome

Overall survival was calculated at five years along with the clinical factors associated with it, and five-year recurrence-free survival was calculated along with the clinical factors associated with it.

### 2.6. Statistical Analysis

Continuous measures were described using median and interquartile range. Categorical and especially binary variables were summarized via reporting absolute and relative frequencies. Overall survival and recurrence-free survival were analyzed using the Kaplan–Meier test. Univariate analysis of factors (gender, ASA classification, metastatic disease, tumor dimension, grade, resection margins, and therapy) associated with overall survival and recurrence-free survival was performed using the log-rank test. Due to the size of the cohort, multivariable analysis was not performed. Statistical significance was set at a cutoff value of 0.05, as usual. Results were generated using the R statistical programming language (R) and IBM SPSS.

## 3. Results

Over the study period, 212 patients were operated on in our department for chest wall resection, amongst which 44 patients presented with primary chest wall sarcoma (Figure 1). The study population included 24 (55%) men and 20 (45%) women, with a median age of 54 years (range 10–87) (Table 1). Upon diagnosis, 13 (30%) patients complained of chest wall pain. Initial tumor assessment revealed the presence of metastases in 10 (30%) patients.

The treatment modalities consisted of surgery alone for 18 (41%) patients, chemoradiotherapy associated with surgery for 10 patients (23%), chemotherapy associated with surgery for 3 (7%) patients, and radiotherapy associated with surgery for 13 (30%) patients. Five (11%) patients did not require a muscle flap or chest wall reconstruction because their tumors were posteriorly situated, located under the scapula (Table 2).

Rigid allografts were used in 4 (9%) patients, and mesh allografts were used in 35 (80%) patients. Only 17 (39%) patients needed to have a muscle flap added by the plastic surgeons to cover the chest wall defect: the latissimus dorsi muscle was used in all these cases (examples in Figure 2 and Figure 3). The post-operative immediate follow-up was marked by complications for 25 (57%) patients; these complications are detailed in Table 2. The CCI^®^ index showed a strong weighting for non-severe complications, with a mean of 8.7 (of 100). No in-hospital deaths occurred. Four patients had to be reoperated on. The causes of reoperations were bleeding, infection, and flap complication. The median hospital stay was 10.5 days (range 5–67).

Table 3 details the histology of the resected sarcoma. Twenty-three (52%) sarcomas were considered low-grade, six (14%) sarcomas were considered intermediate-grade, and fifteen (34%) patients had high-grade sarcomas.

Five-year overall survival was 51.5% (95% CI 36.1%; 73.4%) (Figure 4). Median overall survival was 1973 days (CI 95% 1461; -). As shown in the univariate analysis, the presence of metastasis upon admission and tumor grade were significantly associated with worse survival (*p* = 0.037 and *p* < 0.01, respectively). Local recurrence or resection margins were not significantly associated with overall survival (*p* = 0.59 and *p* = 0.56, respectively). Five-year recurrence-free survival was 71.5% (95% CI 57.6%; 88.7%) (Figure 5). Tumor resection margin status and metastatic disease upon diagnosis were significantly associated with recurrence-free survival (*p* < 0.01 and *p* < 0.01, respectively).

## 4. Discussion

Chest wall sarcomas are a rare and heterogeneous group of malignancies that originate from soft tissues or bones within the chest wall. It is believed that they arise from mesenchymal cells of chest wall tissues [11]. Although their etiology is not fully understood, several risk factors have been identified: genetic factors, previous radiation, and a history of chest trauma play important roles in the development of this entity [12,13]. However, the majority of chest wall sarcomas occur sporadically without any identifiable risk factors. Symptoms of chest wall sarcoma vary depending on the size, location, and invasiveness of the tumor, but most patients experience a palpable mass and chest wall pain as their main symptoms. In our study, 30% of patients complained of tumor-related preoperative pain.

In this retrospective study on chest wall sarcoma, overall 5-year survival was 51%. This finding is similar to the results obtained by Wouters et al., who reported five-year overall survival values of 50% and 63% for 44 patients with recurrent chest wall sarcomas and 83 patients with primary chest wall sarcomas, respectively [14]. Ten (23%) patients had metastatic disease upon presentation in our series. Indelicato et al. reported a similar number of patients (*n* = 9 (23%)), while Laskar et al. reported 25 patients (24%) with metastasis, who were not included in the analysis [15,16]. The presence of metastasis upon diagnosis was revealed to be significantly associated with worse overall survival in our series. Multidisciplinary tumor boards play a key role in the management of these patients via defining a multimodality treatment including chemotherapy and/or radiotherapy in order to optimize survival [14]. In the setting of metastatic disease, the local control of each site via surgery, chemotherapy, and/or radiotherapy are key elements to account for before planning a complete resection of the chest wall sarcoma, as recommended by the ESMO guidelines [5]. Tumor grade was another factor associated with overall survival; this has been previously described in other studies [11,17]. It is important to ensure wide resection margins to avoid local recurrence and optimize overall survival [5]. We report a five-year recurrence free survival of 71.5% (95% CI 57.6%; 88.7%), which is similar to what is reported in the literature: Marulli et al. reported a 5-year disease-free survival of 70% [18]. After univariate analysis, Prisciandaro et al. found that local recurrence was associated with poorer survival [19]. Consequently, the current ESMO-guidelines recommend en bloc resection with an R0 margin for local control for soft tissue and visceral sarcomas, without specifying the minimum distance to the resection margin. Most of our patients had resection margins greater than 2 cm. A section margin of 4 cm was shown to nearly double (56% vs. 29%) the 5 year-survival in malignant chest wall tumors compared with a margin of 2 cm [20].

Chest wall reconstruction is a complex procedure, and it requires experienced thoracic surgeons and, frequently, in the case of a larger skin defect or the filling of dead space, plastic surgeons. Currently, evidence-based guidelines for reconstruction modalities do not exist. In the literature, recommendations for reconstruction techniques are given based on the experience of high-volume centers [7,8,14]. The goals of chest wall reconstruction are the protection of internal organs, the restoration of respiratory function, and the prevention of chest wall instability. In a study by Scarnecchia et al. that included 71 patients, it was shown that the stabilization of the chest wall after resection had a direct inverse correlation with acute respiratory complications [21]: respiratory complications were demonstrated for all patients to whom reconstruction was not applied. There is a consensus that the two most important factors involved in decisions regarding thoracic wall reconstruction are location and defect size. Defects larger than 5 cm and oriented in an anterolateral position should be restored to avoid herniation of the lung and pathologic chest wall mechanics [22,23]. Apico-dorsal defects up to 10 cm in size may not require reconstruction due to adequate protection and stability of the scapula and shoulder [24]. In our series, five (11%) patients did not require reconstruction because their tumors were located under the scapula. Various materials and techniques have been described with respect to the reconstruction of chest wall defects. Synthetic, biological, and titanium meshes or plates are commonly used [25]. Nevertheless, the decision of which material and technique to select is left to the experience of the surgeon, as few adequately performed studies exist regarding the principles of chest wall reconstruction [18]. In a literature review by Mesko et al. regarding the surgical management of chest wall sarcomas, it was concluded that no perfect prosthetic material exists for thoracic wall reconstruction and that any type of prosthesis (whether rigid or flexible) can be used successfully by experienced surgeons [26].

The defect coverage of soft tissues and the obliteration of dead space necessitate close collaboration with plastic surgeons who can apply pedicled or free-muscle flaps. At our center, plastic surgeons are consulted during multidisciplinary discussions of therapy for extensive chest wall tumors to plan flap surgery for chest wall reconstruction. The workhorse flap for anterolateral defect coverage on the chest wall is the pedicled myocutaneous latissimus dorsi flap. In our series, the defect coverage of soft tissue was performed with this flap in most cases. For central defects, such as after the resection of the sternum, the pectoralis major muscle is usually suitable for serving as the sole muscle flap or as a myocutaneous flap with an island of skin. The rectus abdominis flap is the alternative for central soft tissue defects or defects in the lower anterior thoracic region [27].

In our series, no perioperative mortality occurred, but morbidity remained high, affecting 57% of patients. However, using the CCI^®^-index, we showed that with a generally low complication index of 8.7 (of 100), the weighting of complications was usually very low. The observed respiratory complication rate of 9% was lower than that described in the literature, which ranges from 11% to 24% [28,29,30]. Early patient mobilization and daily physiotherapy are intended to prevent such complications: to successfully achieve this, pain management must be accounted for, as it is a key aspect of postoperative care, centered around epidural catheter use. In our series, no material removal had to be performed due to prosthetic infection.

The limitations of this study are represented by its retrospective nature, its long time period, and the limited number of patients, which probably prevented us from finding other factors associated with overall survival such as local recurrence. In addition, the effect of radiotherapy and chemotherapy could not be sufficiently explored due to the low numbers in each treatment group. Further details on whether treatment was delivered in the neoadjuvant or adjuvant setting were not collated for this reason.

As we operate in a large thoracic surgery center, we were able to collect 44 cases of chest wall sarcoma in 19 years. However, since chest wall sarcomas have different histologies and are very rare tumors, it is difficult to collect a sufficient number of cases to provide answers to questions about the effect of ancillary therapy or the influence of new drugs outside of surgery. This is evident in the single-institutional and retrospective nature of other publications on this topic [31]. The literature describes a nonuniform use of adjuvant and neoadjuvant chemotherapy for operable soft tissue sarcomas. However, there may be a positive impact on recurrence-free survival and overall survival for high-risk patients [32,33]. Targteted drugs and immunotherapy are playing increasingly large roles in the treatment of advanced metastatic sarcoma. In a recent review conducted by Fleuren et al., antio-angiogenic multiple-receptor tyrosine kinase inhibitors were shown to have a clinical benefit for osteosarcoma and Ewing sarcoma patients [34]. There are promising results for T-cell based therapies, which may offer new improvements on the treatment of sarcomas [35]. Thus, we recognize that it is necessary to perform multinational and large studies for the advancement of the therapy of chest wall sarcomas.

## 5. Conclusions

Early diagnosis and prompt multimodal management centered around surgical resection are crucial in the treatment of chest wall sarcomas to guarantee optimal overall survival. In the setting of large parietal defects, prosthetic reconstruction and soft tissue coverage through collaboration with plastic surgeons are often required to maintain chest wall stability, prevent flail chest, and preserve pulmonary function. Post-operative complication rates remain high but are of low severity in expert centers with perioperative mortality. Large, multinational studies are needed to help define the optimal management approach for these rare tumors.

## Figures and Tables

**Figure 1 cancers-15-05154-f001:**
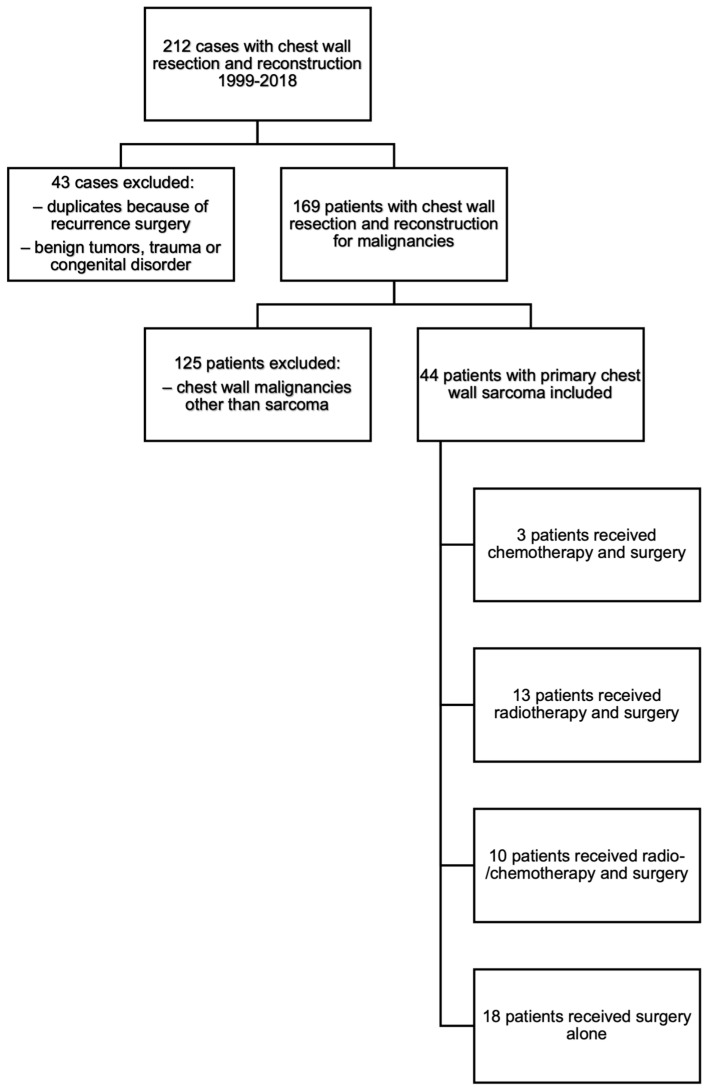
Flowchart.

**Figure 2 cancers-15-05154-f002:**
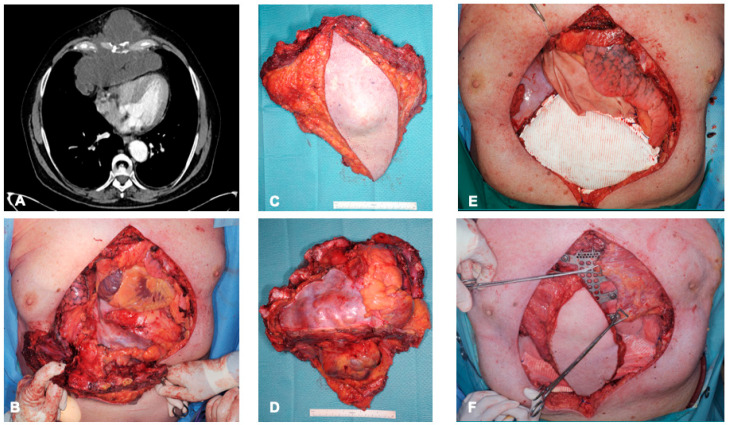
A 68-year-old man with chondrosarcoma of the sternum, with preoperative computer tomography images showing extension into the mediastinum without evidence of cerebral metastases (**A**). En bloc resection involved co-resection of the sternum, pectoralis muscle (bilaterally), anterior pericardium, and central portions of the diaphragm and the transection of ribs 2–6 (**B**–**D**). Layered reconstruction was performed using pericardial patch and a Gore-Tex patch for the diaphragm (**E**). The sternum and rib attachments were reconstructed using custom titanium cage (TRIONYX^®^, NEURO FRANCE Implants, Clamart, France). Soft tissue coverage was performed using myocutaneous latissimus dorsi and left pectoralis flap (**F**).

**Figure 3 cancers-15-05154-f003:**
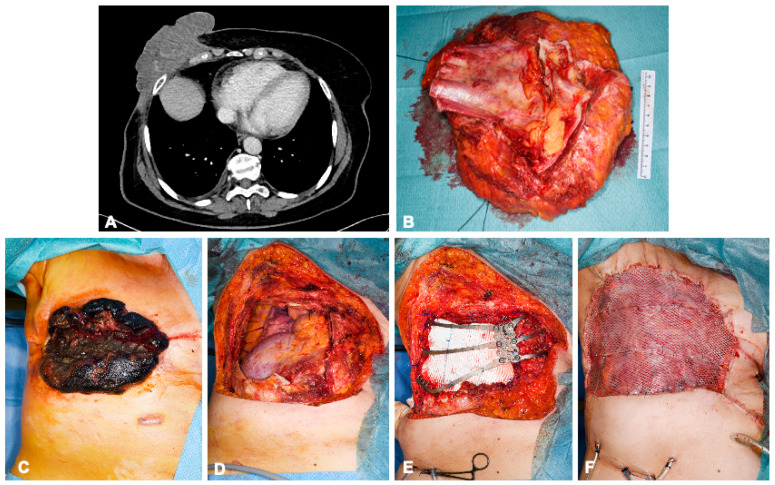
A 68-year-old female patient with angiosarcoma of the right breast 8 years after receiving radiation therapy for breast carcinoma. Preoperative computed tomography images show involvement of the right thoracic wall with infiltration of the pectoralis major muscle, serratus anterior muscle, and intercostal muscles without distant metastasis (**A**). En bloc resection of the exophytic tumor-involved soft tissue, the corpus sterni, and parts of ribs 5–7 (**B**). Reconstruction of the pleura was performed using a Gore-Tex patch and the chest wall with a customizable titanium cage (TRIONYX^®^, NEURO FRANCE Implants, Clamart, France). The soft tissue defects were covered with a pedicled Latissimus dorsi muscle flap and split skin graft from the right thigh (**C**–**F**).

**Figure 4 cancers-15-05154-f004:**
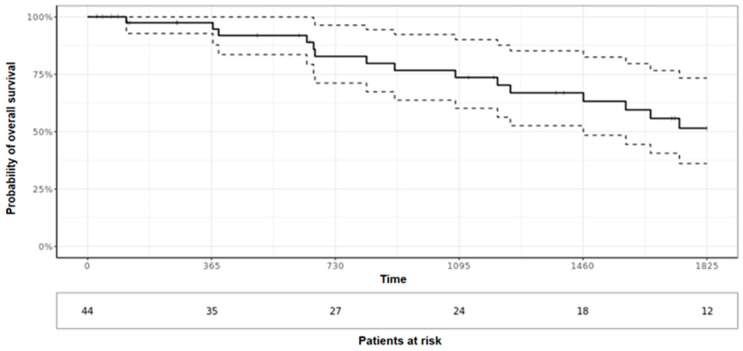
Overall 5-year survival.

**Figure 5 cancers-15-05154-f005:**
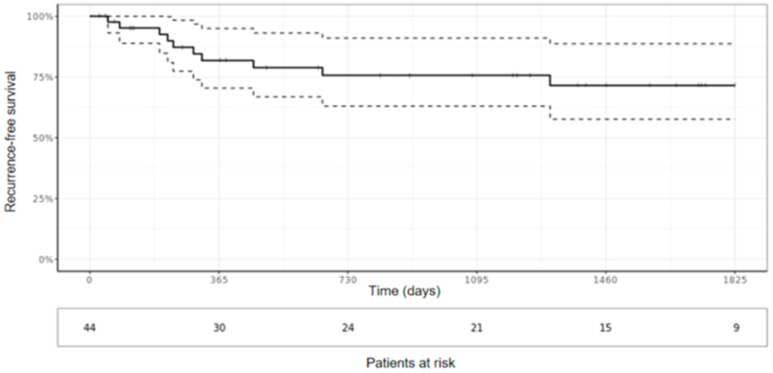
Recurrence-free survival.

**Table 1 cancers-15-05154-t001:** Demographic characteristics.

	Value (%)
Total	44 (100)
Gender	
- Male	24 (55)
- Female	20 (45)
Median Age	54 years (range 10–87)
Median BMI	25 kg/m^2^ range (17–35)
Symptom (pain)	13 (30)
Past medical history	
- Diabetes mellitus	1 (2)
- Pulmonary condition (e.g., chronic obstructive pulmonary disease, emphysema, etc.)	5 (11)
- Cardiovascular condition (e.g., coronary, hypertension, etc.)	10 (23)
Smoker	16 (36)
ASA Classification	
- 1	7 (16)
- 2	25 (57)
- 3	8 (18)
- 4	1 (2)
Metastatic at presentation	
- Primary tumor in the bone (e.g., femur, humerus, etc.)	3 (7)
- Lung	3 (7)
- Lymph node	1 (2)
- Others	3 (7)
Median tumor dimension (in cm^3^)	
- <100	25 (57)
- 100–500	14 (32)
- >500	5 (11)

**Table 2 cancers-15-05154-t002:** Perioperative characteristics.

	Value (%)
Radiotherapy alone	13 (30)
Chemoradiotherapy	10 (23)
Chemotherapy alone	3 (7)
Surgery alone	18 (41)
Chest wall reconstruction (5 without reconstruction because of dorsal location)	
- Rigid allograft (MatrixRIB^®^, Titan Cage)	4 (9)
- Mesh allograft (polytetrafluoroethylene, polypropylene)	35 (80)
Flap	
- M. latissimus dorsi	17 (39)
- M. pectoralis major	2 (5)
Resection margins (in cm)	
- <2	15 (34)
- 2–4	19 (43)
- >4	10 (23)
Tumor grade	
- G1	23 (52)
- G2/High G3	21 (48)
Complications	25 (57)
- Cardiac (e.g., atrial arrythmia)	5 (11)
- Respiratory (e.g., effusion, atelectasis)	4 (9)
- Clavien–Dindo Grade I (any deviation of normal postoperative course without the need for treatment)	15 (34)
- Clavien–Dindo Grade II (pharmacological treatment required)	8 (18)
- Clavien–Dindo Grade IIIa (any intervention without the need for general anesthesia)	2 (5)
- Clavien–Dindo Grade IIIb (any intervention performed under general anesthesia)	4 (9)
- Clavien–Dindo Grade IV (life-threatening complication, ICU)	3 (7)
- Clavien–Dindo Grade V (death)	0
Median CCI^®^	8.7 (range 0–99.9)

**Table 3 cancers-15-05154-t003:** Sarcoma subtypes.

	Value (%)
Bone sarcoma	
- Chondrosarcoma	11 (25)
- Ewing’s Sarcoma	2 (5)
- Osteosarcoma	2 (5)
Soft tissue sarcoma	
- Liposarcoma	8 (18)
- Synovial cell sarcoma	2 (5)
- Leiomyosarcoma	1 (2)
- Myxofibrosarcoma	4 (9)
- Malignant peripheral nerve sheath tumor (MPNST)	3 (7)
- Angiosarcoma	2 (5)
- Undifferentiated pleomorphic sarcoma (former: malignant fibrous histiocytoma)	2 (5)
- Myofibroelastic sarcoma	1 (2)
- Basosquamous sarcoma	1 (2)
- Desmoid fibromatosis	5 (11)

## Data Availability

The data presented in this study are available on request from the corresponding author. The data are not publicly available due to ethical restrictions.

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
