# Peer review of "Outcome Analysis of Treatment Modalities for Thoracic Sarcomas"

_cancers, 2023, doi:10.3390/cancers15215154_

Round 1
Reviewer 1 Report
Comments and Suggestions for Authors
The paper reports the esperiences of one of the leader group on toracic surgery in Europe.
Chest wall sarcomas are rare tumors with different histology and it si difficult to collect a consecutive number of those mesenchymal tumors with the same histology.
This situation is clearly evident even in this paper.
As matter of fact the Authors report 44 cases of chest wall sarcomas met in in 19 years of activity.
Moreover 14 different types of sarcomas are grouped in the paper : some of them are of bone origin, other from soft tissue of the chest wall.
Few are chemosensitive, other exclusively surgery prone.
Radiotherapy was given in 13 patients, but no results are available about adjuvant effects.
In conclusion the Authors confirme that till now only wide surgery can be considered as cornerstone of the treatment and the role of the ancillary treatment cannot be clearly defined.
Only large, multinational, cooperative studies can answer the strategic problems on these specific sarcomas:
1) Is preoperative chemotherapy useful to reduce the resection?
2) Can neoadjuvant chemotherapy improve the not exciting results in PFS and OS?
3) Is there any role for adjuvant treatment?
4) What is the role of new drugs in the treatment of the metastatic/ locally advanced forms?
Author Response
Dear Reviewer,
thank you for your much appreciated comment. We acknowledge your legitimate criticism and have added your remarks to our introduction, discussion and conclusion: As a large thoracic surgery center, we were able to collect 44 cases of chest wall sarcoma in 19 years. Radiotherapy is typically indicated for high-risk soft-tissue sarcomas - Grade 2-3- with neoadjuvant radiotherapy used more frequently in recent years, extrapolating from the data for extremity sarcomas (ESMO Guidelinies, OSullivan et al, Ref 5 and 6). However, since chest wall sarcomas have different histologies and are very rare tumors, it is difficult to collect a sufficient number of cases to provides answers to questions about the effect of ancillary therapy or the influence of new drugs besides surgery. This is evident in the single institutional and retrospective nature of other publications on this topic (Oksuz et al, Ref 31). In our series, the effect of radiotherapy and chemotherapy could not be sufficiently explored due to low numbers in each treatment group. Further details on whether treatment was delivered in the neoadjuvant or adjuvant setting were not collated for this reason. The literature describes nonuniform use of adjuvant and neoadjuvant chemotherapy in operable soft tissue sarcomas. However, there may be a positive impact on recurrence free survival and overall survival for high risk patients (Woll etal, Gronchi et al Ref 32 and 33). Targteted drugs and immunotherapy are playing an increasing role in the treatment of advanced metastatic sarcoma. In a recent review from Fleuren et al, antio-angiogenic multiple receptor tyrosine kinase inhibitors have shown a clinical benefit for osteosarcoma and Ewing sarcoma patients (Fleuren et al Ref 34). There are promising results for T-cell based therapies which may offer new improvement for the treatment of sarcomas (Chew at al Ref 35). Thus, we recognize that it is necessary to perform multinational and large studies for the advancement of the therapy of chest wall sarcomas.
In hope to have been able to serve you with these answers and adjustments in our manuscript, I remain with kind regards
Milos Sarvan
Reviewer 2 Report
Comments and Suggestions for Authors
This manuscript is written compactly,including the own results,which are overall in line with the essential data from the relevant literature.The literature cited is competently selected.The clinical process of diagnostics,multimodal therapy and follow-up care including complication management is comprehensively described.The entire article contains important information on the topic and is easy to read.I recommend the publication in its present form.
Author Response
Dear Reviewer,
thank you for your valued evaluation and recommendation on the publication.
With kinds regards
Reviewer 3 Report
Comments and Suggestions for Authors
Dear Editor and Authors,
Thank you for asking me to review this manuscript titled “Outcome analysis of treatment modalities for thoracic sarcomas” by Dr. Sarvan and his colleagues from the University Hospital of Zurich and the Department of Thoracic Surgery of Dr. Opitz.
In this retrospective, single institution analysis spanning 19 years the authors report their outcomes in 44 patients operated on for primary chest wall sarcomas. All patients underwent major resection and reconstruction. Overall 5-year survival was about 51% which is a quite a good outcome rate considering the pathology.
This is quite a well conducted analysis/report albeit retrospective with although a small number of patients acceptable for the rarity of the pathology reported! It is well written and reported with clear tables and grafts.
Overall I only have one minor comment to ask, specifically:
1. How were patient data obtained? Was there a review of patient medical electronic records/charts or a mining of a dedicated research database? Where there any missing variables/data considering the length of patient recruitment?
Apart from that, I would be happy to accept this work for publication as is. Wishing all the best.
Author Response
Dear Reviewer, thank you very much for your much valued reply. Patient data were obtained from our hospital's internal electronic medical record system and there were no missing variables. With kind regards